# Therapeutic In Situ Cancer Vaccine Using Pulsed Stereotactic Body Radiotherapy—A Translational Model

**DOI:** 10.3390/vaccines12010007

**Published:** 2023-12-20

**Authors:** Kumara Swamy

**Affiliations:** Apollo Hospitals, Bangalore 560076, India; raswamy2002@gmail.com

**Keywords:** in situ vaccination, immunotherapy, stereotactic body radiotherapy, SBRT, pulsed cyclical ISVRT, vascular normalization, CT, antiangiogenics

## Abstract

Both radiation and cancer therapeutic vaccine research are more than 100 years old, and their potential is likely underexplored. Antiangiogenics, nanoparticle targeting, and immune modulators are some other established anticancer therapies. In the meantime, immunotherapy usage is gaining momentum in clinical applications. This article proposes the concept of a pulsed/intermittent/cyclical endothelial-sparing single-dose in situ vaccination (ISVRT) schedule distinguishable from the standard therapeutic stereotactic body radiotherapy (SBRT) and stereotactic radiosurgery (SRS) plans. This ISVRT schedule can repeatedly generate tumor-specific neoantigens and epitopes for primary and immune modulation effects, augment supplementary immune enhancement techniques, activate long-term memory cells, avoid extracellular matrix fibrosis, and essentially synchronize with the vascular normalized immunity cycle. The core mechanisms of ISVRT impacting in situ vaccination would be optimizing cascading antigenicity and adjuvanticity. The present proposed hypothesis can be validated using the algorithm presented. The indications for the proposed concept are locally progressing/metastatic cancers that have failed standard therapies. Immunotherapy/targeted therapy, chemotherapy, antiangiogenics, and vascular–lymphatic normalization are integral to such an approach.

## 1. Introduction

The two primary components of this cancer treatment approach are eliminating the proliferating cells and restoring the immune control mechanism. The first part is carried out by the classical therapies of surgery, radiotherapy (RT), and chemotherapy (CT). The immune mechanism is too complex to be managed entirely by these three methods. There have been encouraging improvements in the cure over time, but a significant population has still developed a recurrence, and the sense of cancer being a fatal disease persists. Immunotherapy (IMT) has changed immunomodulation strategies considerably, along with the rediscovery of the role of RT in immunomodulation.

Both radiation and cancer therapeutic vaccine research are more than 100 years old. The immunological potential of radiation is yet to be explored. Overall, the impression of cancer vaccines as a failure around the turn of the century has, at best, become a strategy with variable outcomes in 2023 [1,2]. There are two root causes for the lack of progress on the immunological front. One is intrinsic heterogeneity and the repopulation of resistant clones [3]. The other mechanism is the “rewiring” of inexhaustible evolutionary web-like pathways of immune escape mutations. There are two types of resistance—primary and secondary. Primary resistance to therapies is a consequence of genomic complexity. Secondary adaptive resistance following therapy results from the poorly understood rapid dynamic rewiring of the transcriptional network of complex crosstalk feedback, leading to survival by adapting to the presence of a drug/therapy [4]. In both the primary and secondary resistance processes, in many patients, either in the short term or long term, the tumor mass transforms into resistant clonal types in treatment-failed patients. Given the dynamicity of cancer cells, cancer recurs due to evolving mutations leading to a therapy-resistant population.

Therefore, in the long term, the answer to battling cancer is to have a durable and consistent method to empower the innate immune network to constantly engage aberrant cells and eliminate them at the first sign of a repopulation of mutant cells. A potential way to achieve this could be by following the vaccination strategy for infectious diseases through repeated antigenic exposures to generate cancer-specific memory cells. In vitro vaccination technology for cancer has come a long way. Yet, its limitation is the continuous in situ mutation of cancer cells, overcoming the effectiveness of such an approach. In addition to the ability to eliminate a tumor mass, RT simultaneously generates tumor-specific antigens and invokes adaptive immunity, making it an “in situ vaccine” initiator [5]. Several studies have demonstrated the effectiveness of in situ vaccination procedures through the intratumoral administration of drugs [6]. However, stereotactic body radiotherapy (SBRT) is the other method that directly causes significant cell lysis, invokes in vivo immunity via immunological cancer cell death (ICD), and releases tumor-specific antigens (neoantigens) and neoepitopes [7]. SBRT has the advantages of precision, technological adaptability to changing tumor morphological–phenotypic changes, and a non-interventional treatment approach. These advantages make SBRT convenient for repeated imaging evaluations, therapy settings, and booster effects through neoantigen generations of altered mutations or subpopulations of cancer cells. SBRT can also be combined synergistically with any in vitro or other in vivo therapeutic cancer vaccine method. However, neoantigens released in vivo are diluted by many nonmutant peptides, unlike molecularly defined vaccines [8]. These nonspecific antigens can cause a “drowning effect”, which may be overcome by appropriate immune adjuvants.

The following review is a compilation of studies on SBRT’s in situ vaccine extractor capability with proposed strategies to optimize it. The SBRT cancer cell lysis and professional phagocytosis of lysed cells lead to the secondary effects of reduced interstitial pressure, improved lymphatic drainage for the process of presenting neoantigens, and improved oxygenation with accompanying vascular normalization. Consequently, there is a shift towards tumor microenvironment (TME)-favoring normal immunological interactions. The present article combines concepts discussed in the earlier published articles to focus on an algorithm for the workable validation and implementation of in situ therapeutic vaccination concepts using pulsed SBRT [9,10,11]. The current review and proposed hypothesis paper specifically compile SBRT as an in vivo anticancer vaccination tool when used intermittently/cyclically in a pulsed manner, along with other immunotherapies.

The radioimmunology perspective is the central theme of this article, focusing primarily on in situ vaccination. Cancer therapy has always been about combination approaches, countering the disadvantage of one by the other’s synergism. Hence, the importance lies in considering all the factors, apparently diverse from each other, to form an integrated, cohesive approach.

## 2. The Proposed Translational Hypothesis

### 2.1. Definition and Components of the Proposed Hypothesis

The classical RT does of 1.8 to 2 Gy per fraction has been used, especially in its formative years, in various fractionation schedules of ascending–descending intensity, concomitant boost, sequential boost, and hypofractionation schedules. All these strategies were primarily directed at overcoming hypoxic/anoxic and resistant populations of cancer cells. The role of SBRT as a potential method for initiating the immunostimulant milieu, the bystander effect, and a possible abscopal response is promising. All these effects are due to SBRT’s ability to produce a significant volume of tumor-specific antigens, which initiates the production of a cancer therapeutic vaccine in vivo. Additionally, radiation is known to act synergistically with immunomodulators, and its use in combination with immunotherapy is gaining momentum [12]. Until now, SBRT plus immunotherapy in clinical use has followed a conventional one-to-five fractions schedule of SBRT. Although the results have been encouraging, they are yet to show a dramatic change in the outcomes of advanced cancer. Most studies have highlighted that using the endothelial-sparing vascular non-disruptive SBRT dose of 5–10 Gy is vital for obtaining optimum effects [13]. Throughout this article, the term SBRT refers to the routine SBRT dose schedule of one to five sessions that exists in clinical practice today, and ISVRT is used when explaining the divided dose schedule of SBRT for the proposed hypothesis.

The proposed hypothesis has two major elements. The first element is that the standard course of SBRT, as routinely practiced today, when split into single doses and given as a pulsed approach, generates repeated pulses of evolving contemporary mutational tumor-specific neoantigens/neoepitopes, leading to primary and booster effects. Consequently, matching priming conditions of antigen-presenting cells (APCs) and T cells occur dynamically (fulfilling one of the fundamentals of vaccination boost doses). This ISVRT pulse dose delivery can be before each cycle of three weekly immunotherapy +/− chemotherapy cycles. This intermittent in situ vaccination RT (ISVRT) schedule and supporting therapies continually improve the tumor’s vascular, immune, phenotypic, metabolic, and mechanical conditions with each cycle. The ISVRT in vivo therapeutic vaccination strategy is also expected to eliminate resistant phenotypic cancer cells as and when they emerge adaptively, enhance the abscopal cascade, and establish effective anticancer immune memory cells. This premise contrasts with the presently clinically established SBRT therapeutic schedule of giving one to eight fractions as a single course, exhausting the potential of a one-course, single-time enhancement of arming antigen-presenting cells. Figure 1 depicts the critical components of ISVRT. Firstly, vascular and lymphatic vessel function restoration are sine qua non for initiating effective vaccine generation. Improved oxygenation, in turn, influences the response to ISVRT. Secondly, following the initiation of ISVRT, there will be incremental interdependent improvement along the course of therapy (as shown in the left side box of Figure 1). Thirdly, adjuvanticity and antigenicity will be enhanced with various available synergistic methods, as shown in Figure 1, Figure 2 and Figure 3.

The second element is that understanding the complexity of the antitumor immune response [14] is essential to designing any strategy. ISVRT effects begin with the creation/extraction of neoantigens of intracellular origin by effective cancer cell lysis and decreasing the interstitial pressure (ISP). The process continues with neoantigen incorporation into APCs; the presentation of neoantigen by APCs to T lymphocytes (depending on various immune-promoting cells and pathways); the activation of T effector cells; and the infiltration of these activated cytotoxic T cells, as tumor-infiltrating lymphocytes (TILs), into the TME (influenced by the vascular integrity, ISP, and the elimination of the immunosuppressive cell lineage in the TME). In other words, ISVRT enhances all steps of the immunity cycle [15]. The endpoint of the strategy is a consistent, effective interaction between T cytotoxic CD8+ cells and cancer cells mediated via the T-cell receptor (TCR) and major histocompatibility (MHC)-type I molecules or tumor epitopes [16]. A deficiency in any of these steps leads to ineffective therapy and immune escape, as well as the recurrence/progression of the disease. This article proposes that effective in situ vaccine creation is a complex process, and it requires not only pulsed ISVRT delivery but also the adoption of various supporting concepts, as detailed below, to have a consistent, definitive, effective, and long-lasting in situ vaccination effect (Figure 2).

Subsequently, continued immunotherapy/chemotherapy/targeted therapy or their combinations are expected to have enhanced action due to resensitization and the presence of induced memory cells. Overall, this approach will likely lead to a significant increase in the control rate and applicability of IMT in a broader population. Additionally, the concept brings forth the added advantage of “titrating” the number of fractions of SBRT to limit the toxicities of SBRT and IMT combinations, where some long-term effects are still unknown.

### 2.2. Evidence for the Proposed Hypothesis

#### 2.2.1. Limited Clinical Data

Almost no clinical data in the literature exist regarding administering RT intermittently. Prolonging the period of RT delivery (technically considered an overall time) beyond a certain period, with the conventional dose being 1.8 to 2 Gy per fraction, results in inferior outcomes in radical treatment approaches [17]. In certain situations, a weekly schedule was practiced in the selected population group in order to reduce the number of hospital visits. The weekly sessions were similar in survival, local control, and acceptable side effects compared to the standard RT used for breast carcinoma [18].

#### 2.2.2. Pulsed RT Approach—Preclinical Studies

Sezen et al. [19] hypothesized that when given in pulses, stereotactic radiation would increase the repertoire of thymus lymphocyte (T) cell priming, incrementally building up immune memory cells against cancer cells. The authors used a combination of immunotherapy and pulsed stereotactic radiation in 12 Gy × 2 fractions scheduled two weeks apart in a lung adenocarcinoma murine model. The results showed enhanced antitumor efficacy in the primary tumor sites, delayed tumor growth, and improved survival in the combination treatment group. Additionally, the pulsed treatment group showed an increased number of cluster differentiation 4 (CD4+) effector memory cells compared to the single-cycle RT group, highlighting the efficacy of targeting multiple points for tumor immune evasion. The authors also suggest that pulsed RT has advantages for multiple-metastatic situations, improving efficacy and safety [19]. Moore et al. also found that pulsed-type radiation dosing in a mouse model was more effective in combination with single-agent immunotherapy in the preclinical model than the traditional daily fractions [20]. He et al. hypothesized that a single dose of RT delivered to a limited number of metastases is rarely sufficient for systemic control, and several rounds of RT akin to IMT cycles have a greater probability of amplifying the adaptive immune response. There was a positive impact on T cell repertoire, high-affinity antibodies, and memory cells. Based on the results of their study, the authors proposed a pulsed schedule with three RT cycles given a month apart to two to four lesions [21].

## 3. Basis for the Proposed Hypothesis

The inability of immune cells to eliminate cancer cells, with a variable accumulation of genetic alterations overcoming the process of “immune editing”, is the beginning of the progression of cancer. The routine regulatory process for preventing cancer manifestation involves a series of carefully orchestrated steps, that is, a group of functions, i.e., the immunity cycle discussed above.

Although RT has traditionally been considered immunosuppressive, the role of the immune-stimulatory effect of RT is increasingly coming into focus. Unlike the usual apoptotic cell death, the radiation effects of acute inflammation and, at a particular dose level, a unique functional immunogenic cell death (ICD) lay the foundation for ISVRT. Also, the demonstration of T-lymphocytes (T)-cell-mediated out-of-the-field tumor response (abscopal response) and the activation of intratumor dendritic cells (DCs) [22] have made radiation a potential in situ vaccination technique. SBRT is a well-established therapeutic modality with clear indications in various types of cancer. Its uniqueness comes from its precision in treating only measurable tumors (with recommended margins around), avoiding significant volumes of normal tissues, and its ability to precipitate cancer cell lysis considered resistant to conventional radiation. Cancer cell lysis releases a variety of tumor-specific antigens (neoantigens), which trigger the initiation of the above-described immunity cycle, making SBRT a convenient non-interventional approach for an in situ therapeutic vaccination method. As discussed above, SBRT impacts multiple steps of the immunity cycle, as detailed in Figure 2. The objective is to harvest the changing spectrum of mutating neoantigens/neoepitopes for improved adaptive and innate therapeutic immunization. Also, ISVRT “unmasks” and “rewires” the mutational pathways de novo, sustaining the immunotherapy response and invoking/enhancing innate immune mechanisms when combined.

For an effective therapeutic vaccine for cancer, some challenges include selecting appropriate tumor antigens and avoiding antigenic drift and “immune-editing”. In some ways, multivalent vaccines are logical. Yet, such vaccines do not obviate the problem of targeting MHC class II epitopes to produce protective Cluster of Differentiation (CD)8+ responses. The other issue is how best to deliver these vaccines to the tumor cells, and animal models have limitations in validating immunotherapies thoroughly, considering their life span. In this scenario, the naturally occurring source of cancer-associated antigens has advantages [15]. Hence, exploring SBRT combinations to their full immunogenic potential is essential after having observed the cancer response away from the treated areas (abscopal effect), albeit in infrequent situations. The present hypothesis proposes a pulsed SBRT dose schedule (ISVRT) to facilitate repeated neoantigen and neoepitope generation to have sustained and prolonged immunogenic effects.

### 3.1. SBRT-Induced Immunity Cycle

A neoantigen is a cancer-induced, unique mutational short-peptide sequence-specific protein. Neoantigens lead to the presentation of Major Histocompatibility Class I (MHC I) molecules on the surface of cancer cells, aiding in distinguishing them from normal cells. CD8+ lymphocytes identify and eliminate cancers when endogenously cytosol-synthesized antigens are presented to the former as peptides bound to MHC I molecules [15]. The neoepitope is a self-defined nine-to-ten-amino-acid sequence frame that can fit into a cell surface Major Histocompatibility Complex (MHC) molecule and a Human Leukocyte Antigen (HLA). Neoepitopes, which are linked to the tumor mutational burden, correlate with a response to immunotherapy [23]. MHC I is ubiquitous, and MHC II is expressed in inflammatory-responding B cells, monocytes, macrophages, and dendritic cells. Antigens ingested into macrophages, dendritic cells, or B cells in endocytic compartments are presented to CD4+ T cells as peptides bound to MHC II molecules for recognition [24].

#### 3.1.1. SBRT In Situ Mechanism of Actions

There are two types of tumor antigens. Tumor-specific antigens (TSAs) are either oncovirus antigens or neoantigens of genomic mutations produced by cancer cells, and tumor-associated antigens (TAAs) are produced by cancer cells and healthy cells. TSAs are related to the total number of mutations per analyzed tumor genomic region, defining the total mutational burden (TMB). There will be a weak response to IMT if the TMB is low. TSAs generate a tumor response, and TAAs can induce “on-target” toxicity reactions within the same structure or “off-target” toxicity reactions in similar organs/structures elsewhere. Most tumor antigens are not/minimally mutated and are considered self-proteins by the immune system. These self-proteins usually “drown” the impact of TSAs. However, despite being overwhelmed by antigens not specific to cancer cells, the significant advantage of in situ neoantigens is the relative ease of their generation during cancer cell lysis [8]. In addition to neoantigens, related immune-enhancing molecules like epitopes are involved in the immune response. Radiation therapy also enhances the generation of neoepitopes and neoantigens, producing an entire repertoire of cluster differentiation 8 (CD8+) T cells [25].

The other critical reversal process is the impact on phenotypic changes. An antigen-specific CD8+ T cell phenotypic analysis shows that radiotherapy enhances antigen-experienced T cells and effector memory T cells. RT activates protein kinase C (PKC) and mitogen-activated protein kinases (MAPK), which are signal transduction pathways as well as transcription factors, making them susceptible to immunotherapy responses via phenotypic changes. Radiotherapy regulates the MHC and increases the expression of specific tumor model antigenic epitope presentation to the MHC on the cell surface, resulting in enhanced amounts of memory phenotypic CD8+ T cells. Radiation can also immunologically facilitate T-cell-mediated cancer cell lysis by altering the biology of the surviving cancer cells [26,27]. Also, SBRT spares the irradiation of the draining lymph node regions, where antigen presentation occurs [27].

Intracellular tumor antigens are presented on the cell surface to induce an antitumor response as MHC molecules interact with the T cell receptors (TCR) on antigen-specific T cells [28]. RT causes gene upregulation, augmented protein degradation, and mammalian or mechanistic targeting of rapamycin-regulated translation, increasing the peptide pool and MHC class I expression. Consequently, more antigenic peptide presentation for recognition happens, ending in an enhanced T cell receptor (TCR)-engineering repertoire. In addition to this induction of tumor antigenicity, especially in immunologically cold tumors, SBRT can cause “adjuvanticity” by invoking regulated immunological cell death (ICD). ICD triggers inflammatory signals through stressed or dying cells, resulting in damage-associated molecular patterns (DAMPs)’ release encompassing adenosine triphosphate (ATP), high mobility group box-1 (HMGB1), and calreticulin. The activation of the stimulator of interferon genes (STING) triggers the transcription of Type I interferons (IFNs) to recruit DCs for maturation, which helps in naïve T cell cross-priming after migrating to the lymph nodes. This adjuvanticity is critical for bridging the adaptive and innate cell responses [29].

#### 3.1.2. Single-Dose SBRT Immunological Effects

Two primary effects of immunogenic cell death (ICD), inflammatory cascades and phenotypic changes, help SBRT extract the needed neoantigens/neoepitopes. ICD can potentially eliminate immunosuppressive cells in the TME for an in situ therapeutic vaccine effect. This release of neoantigens leads to dendritic cell (DC) maturation, antigen cross-presentation, and antitumor T cell response diversification. A single-dose fraction eventually acts as an immune activator with initial lymphocyte depletion. Also, type I interferon phenotypic effects peak at an 8-to-12 Gy single fraction through the cyclic GMP-AMP synthetase interferon genes (cGAS/STING) pathway. In addition to inducing a cluster of differentiation (CD8+) T cell responses through radiation-generated neoantigens, RT also stimulates the immune response via ferroptosis, necroptosis, and pyroptosis [30].

Two important specific points were demonstrated by Lussier et al. in their preclinical study of poorly antigenic cell lines not responding to IMT. Doses of radiation that do not cure cancer can induce MHC-I neoepitopes, leading to tumor lysis populations of CD8+ T cells that sensitize low-mutation-burden cold tumors to IMT. The delivery of a neoantigen-generating immunogenic dose of RT (4 to 9 Gy) throughout the tumor population can initially have a subclonal immune response. This response can subsequently amplify the epitopes already generated, resulting in the rejection of even mutationally heterogeneous tumors [31].

Radiation does modulate the tumor microenvironment at low doses (1–4 Gy) [32]. Yet, this aspect is not discussed in the present article, given the main focus on the antigenicity and adjuvanticity of SBRT in the dose per fraction range of six to ten Gy. However, the potential of a higher antigenicity dose per fraction of 12 Gy or more may have to be considered when the clinical applicability of genetic modulations for protecting endothelial cells and simultaneously sensitizing cancer cells is possible [33].

#### 3.1.3. SBRT Immunological Effect of a Pulsed Dose

Fundamentally, delivering a single-dose SBRT in a pulsed/cyclical manner fulfills the primary condition required for a therapeutic vaccine of booster doses. The pulsed dose delivery generates repeated cycles of neoantigens and neoepitopes to APCs to maximize the vaccine effect, as shown by the animal trial data discussed above [19,20,21].

#### 3.1.4. Single-Dose SBRT Palliative Effects

SBRT, in a single dose of 6 to 8 Gy per fraction, was tried in clinical settings for symptomatic pain relief in spinal metastatic lesions [34].

### 3.2. Improved Phagocytosis, Interstitial Pressure, and Lymphatic Vessel Drainage after SBRT

Broadly, significant components of the TME changes in cancer are immune-phenotypic, metabolic, and mechanical, all in the background of disorganized vasculature and lymphatics. The lysis of susceptible cancer cells through SBRT results in decreased interstitial pressure (ISP), encouraging vascular normalization. These improvements sensitize cancer cells for subsequent doses of SBRT. These changes also facilitate increased IMT and other drug penetration when combined with SBRT. These changes impact the response to treatment and the prognosis [35].

#### 3.2.1. Vascular Normalization

Normalizing the vasculature and overcoming hypoxia are essential for cancer responses. Goel et al. have extensively dealt with this topic [35], and the restoration of the vasculature is the sine qua non for the elimination of advanced cancer (Figure 1).

#### 3.2.2. The Mechanical TME

The bottleneck in the immunity cycle initiated by SBRT is the optimum availability of neoantigens to start the process of DC maturation and drainage to the lymph nodes. Primarily, the mechanical TME impacts this process. The mechanical TME is the product of cell mass solid stress, interstitial fluid stress, and extracellular matrix (ECM) stiffness. The last is due to excessive collagen deposition by cancer-associated fibroblasts (CAFs). The deposited actin fibers crosslink, encouraging further collagen deposition. A cycle of focal contractility (stiffening) starts, aided by transforming growth factor-beta (TGF-β) signaling [36]. Avoiding these changes is critical to long-term disease control, even in a palliative setting. A 6 to 10 Gy dose per fraction is least likely to alter the ECM to irreversible fibrosis [13], thus balancing radiation’s contradictory immunosuppressive and immunostimulatory effects.

Reduced interstitial pressure is inevitable with a decrease in surviving cancer cells. The increased phagocytosis of dead cells due to calreticulin acting as an “eat-me” signal further decreases the ISP. A decreased ISP improves oxygenation and lymphatic drainage. Also, radiation-induced interferon, DAMPS encompassing HMGB1 (via toll-like receptor (TLR) 4 activation), ATP, calreticulin, and heat shock proteins (HSPs) facilitate the recruitment and activation of dendritic cells [37,38], which transit from immaturity to maturity as they migrate to lymph nodes after picking up neoantigens from the interstitial compartment. Thus, this simple step of an effective decrease in the interstitial pressure can potentially exert a potent response, impacting all further phases of the immunity cycle. DCs are the most effective APCs, activating T lymphocytes in the lymph nodes. In the meantime, the cGAS-STING pathway senses damaged nuclear DNA in the cytosol to produce Type I IFN, facilitating DC maturation, the migration of APCs, and T cell cross-priming, increasing the antitumor CD8+ T-cell response. Being a step ahead in the immunity cycle, RT promotes more expression of MHC-1 molecules in murine models, playing a role in cross-presentation and T-cell priming [38]. The use of drugs facilitating macrophage-mediated professional phagocytosis, e.g., oligonucleotide cytosine phosphate guanine (CpG), will reduce the interstitial pressure [39]. If the programmed cell death ligand (PD-L)1 is a “don’t find me signal”, CD47 is a “don’t eat me signal”, and drugs like metformin and small-molecule gefitinib are CD47 suppressants. Therefore, a PD-L1 and CD47 dual blockade [40,41] is another avenue for improving cancer cell lysis and phagocytosis.

#### 3.2.3. Essentiality of Functional Lymphatics (Not Just the UnCollapsing of the Lymphatics)

Compressed and functionally deficient intratumoral lymphatic vessels contribute to increased tumor interstitial pressure [42]. Therapy responses can reduce compression on lymphatics, but the functionality of lymphatics with damaged valves is not restored [43]. In mice bearing human ovarian cancer xenografts with aberrant lymphangiogenesis, the inhibition of TGF-β regained lymphatic function [44]. A report showed the acceleration of functional lymphatic regeneration in wound repair [35,45]. Research is warranted in this direction, considering the criticality of lymphatic vessel valve functionality to “push” the neoantigens and APCs to the lymph nodes. Hence, complementing the critical effect of SBRT cancer cell lysis and releasing collapsed lymphatic channels, along with drugs for lymphatic functional normalization, are crucial components of initiating the immunity cycle. The suboptimal activation of T cells through insufficient neoantigens leads to anergy due to coinhibitory signals [46], which significant neoantigen release through RT combination therapies can offset.

### 3.3. Arming of APCs

Once APCs are formed, the next steps become a self-sustaining process, ending in the formation of cytotoxic T lymphocytes. The vaccine’s efficacy lies in its ability to activate APCs via pattern recognition receptors (PRRs) through neoantigens, which can critically depend on appropriate adjuvants. An in vitro study showed that inflammatory signals that can cause CD8+ T cell activation cannot substitute for direct PRR priming due to their inability to sustain the survival of CTIL differentiation and their inability to reject tumors, indicating the importance of the ability of PRR priming when choosing immune adjuvants. Differential subsets of APCs need to be critically evaluated for the selection of adjuvants with RT combinations, as shown in the disappointing outcome of CpG in some clinical trials due to the absence of TLR9 toll receptors in human conventional DCs. DCs belonging to CD8α+ subsets with TLR3 lack the expression of TLR7, affecting the choice of adjuvants to be used along with SBRT [47].

### 3.4. APCs—T Cell Interaction—Activation of T Cells

The effective priming of cytotoxic CD8+ T cells takes place through tumor-specific neoantigen recognition when presented by DCs and macrophages in the lymph nodes. DCs play a crucial role, although they are small immune subsets in the lymph nodes [48]. Lymph nodes are the classical sites for T cell priming and activation, although the tumor site is capable of the same [49]. Both APCs and anti-PD-1 IMT activate CD8+ T cells in tumor-draining lymph nodes (TDLN), which subsequently traffic to the TME with the help of specific cytokines during RT combinations [48,50]. Tumor-derived factors like interleukins (IL)-6, TGF-β, prostaglandin-E 2 (PGE2), and vascular endothelial growth factor (VEGF), have significant effects on the TME. These factors suppress DCs and turn an M1-type macrophage into an immunosuppressive M2-type phenotype. Simultaneously, regulatory T cells (Tregs) prevalent in the lymph nodes suppress the cross-presentation of neoantigens [48].

### 3.5. Cytotoxic Lymphocytes (CTLs)’ Formation and Trafficking—The Role of SBRT

Activated T cells, as cytotoxic lymphocytes, have unique expressions of a set of homing chemokine receptors (e.g., CXC receptor 3 (CXCR3) and its ligands CXCL9, CXCL10, and CXCL11), complemented by related chemokines in the TME [46]. SBRT’s TME vascular, mechanical, immune-phenotypic, and metabolic effects will attract the CTLs trafficking to the tumor site, converting the immune “cold nodule” to a “hot nodule”.

### 3.6. SBRT-Assisted In Situ TIL Infiltration

SBRT increases the endothelial expression of adhesion molecules (intercellular adhesion molecule-1, vascular cell adhesion molecule-1, and E selectin) at the tumor site and initiates the adhesion of CTLs. RT-generated IFNγ production within the primary tumor facilitates cytotoxic T-cell trafficking, adhesion to the endothelium, and diapedesis [51]. Vascular normalization plays a critical role in these steps. Vascular normalization by RT improved the efficiency of adoptively transferred cytotoxic T cells in a mouse model [52]. Activated T cell tumor infiltration involves, in addition to chemokines, CXC receptor 3 (CXCR3) and its ligands CXCL9, CXCL10, and CXCL11, as well as CXCL5. At the tumor site, adhesion molecules, including selectin ligands, help bind to blood vessels and extravasate, resulting in TME infiltration. In the next step, cytotoxic CD8+ T cells recognize the target cancer cells through MHC-I molecules and induce T-cell-mediated cytotoxicity [46].

## 4. Factors Mitigating the SBRT In Situ Response

The high-dose schedule of SBRT, while initiating a domino effect of immunostimulation, mainly by releasing neoantigens and neoepitopes, simultaneously invokes several immunonegative effects in the TME. Consensus regarding optimal dose/fractionation schemes and the optimal combination of RT and IMT is yet to be reached [53].

### 4.1. Hypoxia/Anoxia

Since the inception of RT, an accumulation of studies has shown that hypoxia has been a reason for resistance to radiotherapy since the presence of oxygen molecules is essential for the fixation of radiation-induced deoxyribonucleoside (DNA) damage that causes cancer cell death. Additionally, hypoxia leads to immunosuppressive cell types such as Tregs and MDSCs, M2-polarized tumor-associated macrophages (TAMs), helper T cells (TH)2-polarized dendritic cells (DCs), the downregulation of MHC class-I molecules and natural killer (NK) cell-activating ligands, and the upregulation of PD-L1 on tumor cells that combinatorial immunotherapy and SBRT can counter. Thus, hypoxia is responsible for immune tolerance via multiple mechanisms [54].

Normoxia is essential for the initiation and completion of the immunity cycle. In intensely immunosuppressive, intrinsically radioresistant hypoxic tumors, reoxygenation with fractionated SBRT in combination with IMT enhances local control and abscopal effects [55]. A single dose of ~8–10 Gy is the lower threshold for significant endothelial apoptosis [56]. A higher dose per fraction would cause aggravation of hypoxia, converting a durable response to a short-term control [33].

#### Overcoming Hypoxia

Reoxygenation, radiosensitivity, redistribution, repopulation, and repair have been fundamental to the evolution of modern-day fractionated RT. The first three of these factors lead to the beneficial effects of fractionation, especially improved oxygenation, and the latter two are responsible for the failure of treatment in some patients. The classical understanding of repopulation (with accelerated multiplication) happens around 4 weeks in head and neck cancer [57]. However, it can occur earlier in the third week [58]. The proliferation of these surviving cells, especially stem cells, leads to local relapse [59]. Given the overall effect of reoxygenation and redistribution on sensitive cell cycle phases, fractionated RT became the standard. Hypoxic cells are two to three times more radioresistant than oxic cells, and the theoretical concern of repopulation increasing hypoxia remains [59]. According to the concept of the “reoxygenation utilization rate”, a single fraction cannot utilize the phenomenon of reoxygenation. This lack of time for reoxygenation deprives one of the five radiobiological principles of the ability to increase the efficacy of fractionated radiation. Oxygen utilization increases to about 87% with six to eight fractions, with an enhanced response [60]. Based on these findings, IMT given after SBRT may improve drug delivery, distribution, and acceleration of TIL accumulation in the TME. AAGs, when combined appropriately with the process of vasculature normalization, also improve oxygenation [35].

### 4.2. Immunosuppressive Recoil—Tregs, BMDCs

#### 4.2.1. Importance of Concomitant Immune Tolerance from Untreated Lesions through RT

Zachary S. Morris et al. showed results in their mouse tumor model with two implanted lesions. They treated one lesion with RT plus an intralesional agent. They demonstrated Treg-mediated immune suppression of the primary lesion by the untreated lesion. This feedback immune suppression is reduced by depleting the Tregs, thus restoring the vaccination effect. The authors referred to this phenomenon as “concomitant immune tolerance”. The findings reveal that the untreated lesions exert a “tumor-specific”, Treg-dependent (not ruling out Treg-independent mechanisms) suppressive effect. The feedback immunosuppressive cascade was less in the case of small, untreated lesions [61]. These findings raise a hypothetical question: does this suppressive recoil mechanism extend to the systemic microscopic tumor burden? Therefore, “concomitant immune tolerance” could be one of the reasons for the abscopal effect not living up to its expectations, even when combined with immunotherapy. Morris et al. also demonstrated that immunosuppressive effects were seen in the 12 × 1 Gy fraction and 8 Gy × 3 fractions, even though the latter is considered more immunogenic [61], indicating the complex nature of the immune playout.

Immunosuppressive bone-marrow-derived cells (BMDCs), with the first influx happening within 3 to 5 days and the second wave after two weeks, are the other mitigators of the systemic and local effects of SBRT. The mobilization of BMDCs is more than double with 15 Gy, compared to 8 Gy per fraction [13].

#### 4.2.2. Overcoming Tregs and BMDCs

Treg cells, essential for normal immune homeostasis, have several immune suppressive mechanisms. Immune checkpoints, chemokines, and small molecules have been used to target Tregs. However, the results have been variable and left a “limited therapeutic window”, with the issue of Treg cell depletion leading to autoimmune adverse effects [62]. Treg depletion for a limited period by anti-CTLA-4 monoclonal antibodies (mAb) is sufficient for activating CD8+ T cells with an antitumor response despite the quick recovery of Tregs from depletion [63]. The strategy inducing Treg depletion for a short period, starting with RT +/− immunotherapy, reduces the risk of adverse autoimmune events.

The other engaging approach is cyclophosphamide in lower metronomic doses that can selectively cause “sensitive” Treg cell apoptosis. However, adding cyclophosphamide to standard-care sunitinib combinations in renal cell carcinoma failed to show survival benefits. Using pulsed SBRT and low-dose cyclophosphamide is an option to reduce the concomitant immune tolerance initiated by Tregs. The tyrosine kinase inhibitors (TKIs) imatinib and dasatinib inhibit Treg cell function, causing Treg cell reduction in chronic myeloid leukemia. Blocking the VEGF and vascular endothelial growth factor receptor (VEGFR)2 axes is another potential approach [62].

In summary, a short course of anti-CTLA-4 mAb immediately after low-dose cyclophosphamide and multitarget TKIs looks promising to counter the phenomenon of concurrent immunotolerance, with the added advantage of TKIs’ role in vascular normalization and decreasing hypoxia. The tumor-promoting two-peak influx of BMDCs is effectively blocked by stromal cell-derived factor 1 (SDF-1)/chemokine receptor 4 (CXCR4) when given immediately after SBRT [13].

### 4.3. Activation of the Transcription and Export (TREX)1 Pathway

A higher dose per fraction of SBRT (>12 Gy per fraction) increases the expression of the three-prime repair exonucleases gene (TREX1), which reduces the accumulation of cytoplasmic dsDNA, impacting IFN-ϒ production and causing TME immunosuppression [29]. RT at lower doses propagates DC maturation and immune cell priming, as Demaria et al. showed. Some studies showed better tumor control with a 15 Gy single dose [29,52].

#### Preventing TREX1 Pathway Recoil

The 6-to-10 Gy dose range is effective for reducing TREX1 pathway activation and optimizing the creation of neoantigens/neoepitopes.

### 4.4. Other Immunosuppressive Recoil Pathways

DCs can cause immunosuppression concurrently by causing a shift from Th 1 to Th 2, mediated by a CD8+DC-inducing cytokine tyrosine kinase-3 ligand (FLT3 ligand). The RT-generated signal transducer and activator transcription 3 (STAT3), a regulator of the forkhead box protein (FOXP) 3 gene, converts CD4+ cells to immunosuppressive Treg cells, the inhibition of which leads to a decrease in the number of Tregs, MDSCs, and M2-phenotype macrophages in the TME [38].

Radiation-induced cluster differentiation-47 (CD47) downregulation has a DC phagocytosis suppression effect [29]. One function of the binary complex in autophagy is the degradation of MHC I into T cells and DCs, MHC II into MDSCs, and impaired PD-L1 degradation, all of which cause deficient neoantigen presentation and impaired T cell action [14], which needs further understanding.

#### Overcoming Other Immunosuppressive Pathways

A significant transforming growth factor beta (TGF-β)-causing immune-hostile TME with 12 Gy per dose has a minor impact when a 6 Gy dose of SBRT is delivered [13]. The antihistamine and antifibrotic drug tranilast is a TGF-β inhibitor that can assist in the suppleness of the extracellular matrix (ECM) for a cancer-cell-inhibitory TME. Combining the monoclonal DC101 (VEGFR antibody) and an anti-TGF-b1 antibody helps normalize and reduce collagen density. There is unlikely to be a clinical benefit to targeting fibroblastic activation protein (FAP) alone in the TME [64].

Immunogenic cell death (ICD), a hallmark of the SBRT dose schedule, is a type of cell death that releases damage-associated molecular patterns (DAMPs) and ATP and causes the translocation of the ER chaperone calreticulin, which in turn is responsible for recruiting DCs and phagocytic macrophages. However, the rapid accumulation of the catabolic product of ATP (adenosine) suppresses DCs, increases Tregs, and favors the transition of TAMs to the M2 phenotype through TGF-β. These changes can be countered by a bifunctional fusion protein, which can block both the PD-L1 and TGF-β pathways [64].

### 4.5. Summary of Overcoming the Immunosuppressive Pathways

The sustained improvement of hypoxia through the normalization of the vasculature gives rise to a cascading immune-favorable TME [13]. Limiting the SBRT dose per fraction to <10 Gy in itself mitigates several immunosuppressive pathways, like the activation of the TREX1 pathway [29]. Even Treg activation becomes reduced with an SBRT dose schedule of <10 Gy per fraction [46]. Additionally, several other strategies can be adopted to convert a suppressive TME to an immunogenic TME: 1. the strategic use of AAGs for vascular normalization; 2. the use of a TGF-β blocker to restore the function of lymphatics and to keep the ECM supple; 3. the use of Treg and MDSC cell spike suppressors (with minimal periods of Treg depletion); and 4. the use of nanomedicines to hasten the “wheel of immunity cycle”, potentially by several steps.

## 5. Subclinical Disease, Dormancy, ECM Suppleness, Memory Cell Cycle, and the Abscopal Cycle

### 5.1. Memory Cell Cycle

With a reduction in the tumor load to the subclinical level of <10^5^ cancer cells, immune surveillance is likely to be restored. This condition sets the subclinical cycle of immune editing, equilibrium, elimination, or immune escape with re-entry into the immunity cycle, depending on the ECM immune profile (“soil”) that the initial therapies create. RT, CT, and IMT can generate persistent, long-lived, and self-renewable memory T cells. The same is suitable for CAR-T cell therapy. APCs, through MHC II molecules, activate CD4+ T cells, which in turn present antigens to B cells, resulting in the secretion of antigen-specific antibodies responsible for the humoral immunity of macrophages, induce antibody-dependent cell-mediated cytotoxicity (ADCC) mediated by NK cells, or end up as memory cells [65]. Claire C. Baniel et al. [66] hypothesized that a combination of RT, intratumoral immune cytokines, and an immune checkpoint blocker as an in situ vaccine primes memory B cells, which are responsible for a persistent humoral response. However, the exact role needs to be clarified [66].

Surgery did not display a rejection of the tumor against the rechallenge in animal models [66], indicating that the sequencing of surgery may need to be rethought. To invoke memory cells, SBRT, with or without other in situ vaccination techniques before surgery, may be more desirable than upfront surgery in certain situations (akin to the present-day management of carcinoma rectum). The most significant advantage of surgery would be eliminating the final group of immune-intransigent cold residues for a cure or long-term control.

Baniel et al. observed an increase in memory-derived antitumor IgG antibodies even before engraftment on rechallenge, indicating antitumor activity even before the macroscopic manifestation of cancer [66]. Thus, to overcome the Treg-propagated immunosuppressive effect of the larger nonindexed lesions [61], SBRT plus or minus intratumoral agents as “vaccination generators” along with checkpoint inhibitors can be used when the clinical extent of the disease does not permit the inclusion of all lesions. The other option would be combining systemic targeted radionuclide therapy alone (to suppress “concomitant immune tolerance”), alternating or in combination with SBRT for an in situ vaccination approach [32].

### 5.2. The Role of NK Cells in Minimal Disease and Dormancy

NK cells are professional killer cells with cytotoxic granules, pore-forming perforin proteins, and granzyme proteases acting on target cell membranes. NK cells are an integrated innate immunity system that removes stressed and transformed cells. They are found to be fewer in number in an established cancer mass and play a significant role in early cancer, minimal metastases, and dormancy stages [67]. NK cells are unique in the cancer surveillance system because of their ability to identify the aberrant feature or loss of MHC-I molecules, a phenomenon of recognizing abnormal cells with “missing self-recognition” [67]. Cancer cells become susceptible to NK cells by evading cytotoxic T cells in certain situations. NK cells can activate a battery of “natural cytotoxicity receptors” on cancer cells. IL-21 and IL-2 exhibited synergistic effects on NK cell activation [67]. In an in vitro study, high-dose ionizing radiation affected NK cell function (a low dose is a stimulator of NK cells). The NK cell function was not impacted by interleukin-2 (IL-2) pretreatment. NK cells can generate antigens for DCs, indirectly contributing to T-cell responses to cancer [68].

## 6. Amplification of the Antigenicity and Adjuvanticity of SBRT

### 6.1. Immunotherapy (+/− CT/AAGs) Complementing the In Situ Vaccination Effect of SBRT

#### 6.1.1. Immunological Evolution of a Cancer Mass

Neoantigens are the product of cancer-induced mutations that ordinarily elicit an immune response but, over time, might provide “fuel to the fire” of cancer evolution en route to immune escape. However, neoantigens can also be presented on cell surfaces, recognizing the cancer cells as “non-self”, leading to lysis, referred to as immune editing. The dynamics of immune editing can lead to fluctuations in the contraction or expansion of the cancer cell population, leading to subclones. Over time, an increased load of neoantigens avoiding immune editing leads to evolution and immune escape [69]. Immunotherapy can cause a variable rate of cell death yet cause severe, recognizable neoantigen depletion, making cells resistant despite the high tumor mutational burden. The clonal escape mutations continue to be “antigen warm” (high neoantigen burden) yet “immunologically cold”, with several subclonal neoantigens. With the rapid shrinkage of the active clonal population, passive clones continue to grow by pruning antigens, turning them into immune-cold tumors. Sometime after immunotherapy, the immune landscape differs from the original tumor, with new clonal emergence requiring the following/additional lines of therapy [69]. Since the total mutational burden is proportional to neoantigen production [70], proliferated subclones can be expected to have reestablished, modified neoantigens.

#### 6.1.2. The Difference That SBRT Can Make

Some consider the reactivation of the antitumor immune response by RT to be the sixth R of radiobiology [71]. SBRT can play a role in all steps of the immunity cycle in combination with other local or systemic therapies. The process can be furthered by increasing the production of tumor-specific antigens by local combination therapies with SBRT, given its ability to act synergistically, bearing in mind the possibility of overlapping short- and long-term toxicities. The process includes enhancing, more specifically, type-1 T helper (Th1) and type-2 T helper (Th2) cells that activate immune stimulatory macrophages and B-cells, respectively. Negating the TME immunosuppressive cycles and stimulating TME immunity pathways [14] requires well-designed combination strategies.

Mutations outside the anchor position interact with the peptide/MHC complex with TCR, while mutations in the anchor position potentially create a high affinity for the recognition of mutated specific neoepitopes. However, vaccination or personalized adoptive cytotoxic T lymphocyte (CTL) transfer treatments favor the evolution of escape variants through a loss of global MHC expression [72]. Systemic natural killer (NK) cell activators and chimeric antigen receptor (CAR) T immune cells may preferentially recognize MHC-negative tumor cells for elimination. The other combination option is surgical excision or cytoreduction for a resistant residual lesion [72].

IMT, on average, has benefits in 12% of patients, indicating a significant amount of primary and secondary resistance, along with the burden of a large proportion of toxicity risk. Resistance can dominate at the TME level even when high levels of antigen-specific T cells are in circulation. The primary problem is poor neoantigen availability with low-TMB tumors, leading to the suboptimal immune priming of APCs. Both types of resistance need to be overcome by the use of immune stimulatory agonists that promote stimulatory interferon genes (STING) and Toll-like receptor (TLR) signaling [29]. The reprogramming of “cold” or even a therapy-resistant “immune-desert” tumor to “immune-hot” tumors can happen just by targeting hypoxia. Hypoxia is the foundation on which immune resistance rests due to countless alterations in metabolites, TME-progressive acidosis, immune checkpoint expression, and cancer-favorable TME immune depletion, as well as hypoxia-induced tumor-promoting autophagy [30], resulting in a plethora of resistant pathways.

Along with RT, additional strategies include interleukin-15 (IL-15) agonism (a known activator of NK and CD8+ cells), augmenting the number of antitumor memory CD8 T cells; the intratumoral depletion of regulatory T (Treg) cells by targeted glucocorticoid-induced, tumor-necrosis-factor-receptor-related (GITR) agonism; and a combination of anti-programmed cell death ligand-1 (PDL-1) and ipilimumab, leading to positive feedback with further destruction of myeloid-derived suppressor cells (MDSCs). Immunogenic cell death caused by RT increases damage-associated molecular patterns (DAMPs), made up of group box-1 proteins (HMGB1), impacting toll-like receptor-4 (TLR-4) (and thus dendritic cells), which facilitates more extensive antigen presentation due to the inhibition of intracellular antigen degradation and adenosine triphosphate (ATP), stimulating the aggregation of dendritic cells for antigen presentation in the tumor [73].

The combination of local radiation with Th1 cell therapy augmented the generation of tumor-specific CTL at the tumor site and induced the complete regression of the tumor, although radiation therapy alone did not exhibit such a pronounced therapeutic effect [74].

#### 6.1.3. Newer Avenues of RT and CAR-T Adoptive Cell Therapy

Systemic natural killer (NK) cell activators and chimeric antigen receptor (CAR) T immune cells may preferentially recognize MHC-negative tumor cells for elimination. [72]. This therapy has surfaced as a breakthrough IMT in cancer, especially in hematological malignancies. However, in solid tumors, antigen escape, toxic reactions, abnormal vascularization-furthering tumor hypoxia, and an insufficient infiltration of CAR-T cells leading to immunosuppression are the limiting factors [75]. The mechanism of action is due to interferon-gamma (INF-γ), tumor necrosis factor-alpha (TNF-α), perforin, and granzyme-induced cell lysis acting by recognizing and binding to proteins or gangliosides on cancer cell surfaces. The interaction is independent of antigen-derived peptides and class I molecules of the MHC complex. RT facilitates CAR-T cell homing by causing TME inflammatory changes, a migration of CAR-T cells following decreased ISP, and reoxygenation. Increasing integrin intercellular adhesion molecules-1 (ICAM-1), vascular cell adhesion molecule-1 (VCAM-1) expressions in endothelial cells, activating complementary endogenous several antigen-specific responses, and converting TME immunosuppressive to immune-promoting cells are other actions of RT. Correspondingly, CAR-T cells sensitize and target radioresistant cells. An in vivo RT dose of as low as two Gy per fraction has been shown to expand CAR-T cells. Yet, another preclinical study on pancreatic cancer found this schedule to be insufficient. In a preclinical glioblastoma study, a 4 Gy × 1 fraction showed the extended survival of CAR-T cell therapy, and a certain number showed complete regression [75]. A pulsed RT schedule would be worth trying along with CAR-T cell therapy.

#### 6.1.4. SBRT–IMT Timing

Timing with IMT is the most critical of all the factors discussed here. RT can lyse the immune cells locally. It can increase immune suppressive cells, signal pathways, and immune suppressive factors [76]. Hence, synchronizing the SBRT and IMT sequences is critical. There is a general understanding that SBRT after IMT is less effective, and the optimum time may be anywhere from 48 h to <7 days [13,77]. IMT administration on days 6–10 after RT was more effective in a mouse model than on days 1–5 or 11–15, according to study results by Morris et al. [61]. Considering the critical aspect of reoxygenation and the initial TME lymphocyte depletion recovery, the interval after SBRT could be at least 48 h [60].

### 6.2. Further Points about Enhancing the Antigenicity and Adjuvanticity of SBRT

#### 6.2.1. Alternative RT Therapies

In addition to IMT +/− CT, there are other systemic approaches to enhancing the antigenicity of SBRT. In HCC, with Yttrium-90 radioembolization along with an increase in APCs, the CD8+/CD4+ T cells in peripheral blood were associated with the upregulation of the chemokine (CCL)5 and CXC-ligand (CXCL)16 pathways, an increase in the infiltration of CD8+ T cells and NK cells into the tumor, and the subsequent activation of dose- and fractionation-dependent immunosuppressive response [78]. The other approach is low-dose RT directed to asymptomatic lesions to balance the “recoil” immunosuppressive effects. Low-dose RT-mediated increased tumor PD L1 expression via the IFNγ signaling peaking at 72 h can be achieved through the concomitant use of anti-PD-1 or anti-PD-L1 [78]. Clinical benefit was absent in colon and lung cancer phase II clinical trials when low-dose/moderate-dose (8 Gy × 3 fractions) RT was combined with a CTLA-4 and PD-L1 dual blockade. This lack of benefit is presumed to be the result of single-site, low-dose RT. Low-dose RT through radionucleotide therapy, targeting all the lesions (visible and invisible on imaging), is the other option and did show better survival when combined with anti-CTLA-4 IMT. The exciting findings with the radionuclide therapy approach were the STING-dependent stimulatory effects on NK cells, the increased CD8+/Treg ratio within a day (decreased Tregs), the increase in T effector cell infiltration, decreased intratumoral MDSCs, decreased IL-10-secreting macrophages, and the reduction in CD8+ T cell exhaustion, even in the TME of poorly immunogenic tumors (compared to LDRT/moderate-dose RT). Combining focal high-dose RT, IMT, and radionuclide therapy led to the best response in both lesions treated with external radiation and lesions where external radiation was not given. LDRT to secondary sites (small volume lesions), combined with moderate/high doses to selected primary sites (large volume lesions) of cancers and the dual blockade of CTLA-4 and PD-1 in vivo, improved the abscopal effect and survival. The study showed that the improvement was CD4+ and NK-cell-dependent. This finding paves the way for combining LDRT, high-dose RT, and IMT in a tri-modality approach [46]. According to the present author, these approaches may be revisited with a pulsed/cyclical moderate-dose RT (6 to 10 Gy per fraction), with the added advantage of control over long-term toxicities.

#### 6.2.2. Potential Combinations of Other In Situ Techniques with SBRT

Several intratumoral injectable adjuvants have been tried as in situ vaccine methods and immunotherapies to improve antigenicity and adjuvanticity. These include the following TLRS agonists: cytosine phosphate guanine (CpG); an analog of synthetic double-strand ribonucleic acid (dsRNA) polyinosinic–polycytidylic acid (Poly-IC); feline sarcoma-like tyrosine kinase-3 ligands (FMS-FLT3L); TLR7/8 agonists; nanoparticles; oncolytic viruses; oncolytic peptides; thermal therapy; stimulator of interferon genes (STING) agonists; melphalan; and rose bengal disodium (PV-10). However, the limitations of these therapies are their accessibility for administration, especially repeated injections, unequal distribution within the tumor [29], and disruption of the vasculature. Selecting an appropriate adjuvant is critical, matching the subsets of DCs involved as APCs in humans, to avoid disappointing clinical results [47]. Some of these disadvantages can be overcome with nanomedicines [65]. The in situ effect can be broadened by logically combining RT with one or more of the techniques mentioned above. The combination of ICD-capable RT with in situ IL-12, avoiding toxicities in the systemic administration of IL-2, is promising. In a clinical trial, the in situ administration of FLT3L and a TLR3 agonist along with RT renewed the checkpoint inhibitor blockade and showed a durable response [78]. The dendritic cell growth factor and tyrosine kinase 3 (Flt3) ligand demonstrated an abscopal effect [5]. Immunologically cold tumors had a ten-fold increase in CD8+ infiltrates with combined intralesional in situ agents and RT therapy, which jumped to 18-fold when anti-cytotoxic T-lymphocyte-associated protein 4 (anti-CTLA-4) was added [61].

#### 6.2.3. Other Local Therapies

The effectiveness of any vaccine against infectious disease is based not only on antigens but also on immune adjuvants. In situ vaccination with the cancer tumor nidus itself supplies the antigen [79]. Immune adjuvants act as slow-release systems in the nidus, ensuring continued stimulation of the immune system [80]. One technology suggested is leading and activating DCs combined with antigens generated from thermal tumor ablation, which later acts as a nidus for slow-release, sustainable immune stimulation for the drainage of lymph nodes. However, there is a possibility of the rebound recovery of cancer cells at the edge in the sublethal temperature zone [80], which is well known to vascular disruptive agents (VDAs) due to the central necrotic area with a viable rim of cancer cells along with aberrant neovascularization, as shown in preclinical models [81]. These trials of VDAs over the decades validate the importance of the preservation of vasculature as a therapeutic principle. However, a combination approach to counter this “rim effect” is ongoing with further trials [81]. Since tumor control with vascular disruption with ablative doses of stereotactic radiosurgery (SRS) and other methods is more likely to be a short-term benefit [33], the timed repeated delivery of RT with a non-vascular-disruptive dose is likely to maintain long-term immune stimulation without the risk of rebound aberrant vascularization and repopulation at the edge of ablation that is seen in the vascular disruptive methods/drugs.

#### 6.2.4. Making Use of Autophagy

The autophagy process has binary roles as an immune promoter or suppressor, depending on the TME. In the immunoprotection role, autophagy interacts with APCs, T-cells, macrophages, MHC I cross-presentation of autophagy-dependent neoantigens, and MHC-I and MHC-II in DCs and creates memory cells [14]. Autophagy can help in the extraction of antigens when lysosomal enzymes isolate and degrade intracellular antigens under the action of the autophagolysosome into several peptides, which are transferred to the APC cell surface to interact with MHC II in the presence of protein molecules such as calreticulin. The induction of the mitophagy of the defective mitochondria pathway in tumor cells that lack a signal transducer and transcription-3 (STAT3) activator leads to enhanced antigen presentation. Also, autophagy-originated, newly developed epitopes improve antigen availability [14].

## 7. Note on Minimizing Toxicities

Since SBRT has the propensity to invoke an inflammatory cascade via DAMPS interacting with pattern recognition receptors (PRR), it can potentially cause severe immediate and long-term side effects. With the increasing use of immunotherapy, dramatic improvement in long-term survivors is possible. Therefore, avoiding fibrosis and maintaining extracellular matrix (ECM) suppleness is as crucial as the cure.

RT can potentially cause long-term severe complications due to progressive senescent changes in endothelial stem cells and the fibrosis of the ECM, preventing immunogenic cross-talk [82]. RT and IMT must be considered a “double-edged sword” that needs careful planning to avoid the synergistic adverse effects because of the higher risk in the combination RT + IMT therapy. The adverse events can be fewer when the interval between RT and IMT is higher with certain IMTs, where both tumor cells and normal tissues develop immune responses rather than cancer cells alone [83].

Ways to reduce the possible toxicities of combined therapy are (a). Limit the dose per fraction to less than 10 Gy and as low as 5 Gy per fraction to avoid vascular endothelial disruption and senescence, which causes fibrosis in the long term. (b). Titration the dose per fraction and the total dose of SBRT. In the event of the appearance of higher-grade acute toxicities, the subsequent doses per fraction can be reduced, or subsequent fractions can be skipped altogether. It can be reintroduced with later IMT cycles in the case of persistent residual lesions if they are not operable. (c). Modulate the RT dose distribution to reduce the dose to the critical structures nearby. (d). Adopt the subvolume treatment techniques described by Tubin et al. [84].

There are five intertwining categories that the above literature review shows. One are the measures and pathways that enhance the impact of the SBRT dose per fraction; the second is mitigating/reversing the immune-suppressive recoil factors with SBRT combinations; the third is overcoming the immune escape by invoking long-term memory cells; the fourth aspect is adding antigenicity and adjuvanticity to the effects of SBRT; and the fifth critical approach is minimizing the toxicities of combination therapies by basically keeping the ECM supple without inflammatory/fibrotic changes.

## 8. Hypothesis Validation Algorithm for Pulsed/Cyclical/Intermittent ISVRT with Supporting Stratagems

RT has been tried in various fractionated schedules, but probably not in chemotherapy/immunotherapy schedules of weekly/3-week cycles. Based on the literature reviewed above, the present hypothesis proposes using the standard SBRT schedule in divided fractions as an ISVRT, where each fraction is delivered before an immunotherapy cycle at a minimum of 2–3-week intervals. Other therapies can be integrated into this ISVRT schedule for improved in situ vaccine effects (Figure 3). The ISVRT cycles can continue until the disappearance of measurable lesions or the appearance of unacceptable acute toxicities (a total of about five pulses of ISVRT). After the ISVRT dose schedule is completed, the immunotherapy combinations can be continued as per the existing protocol. Non-responding tumors before ISVRT are likely to respond with continued courses of IMT combinations. This proposal suggests a single effective dose of RT, not necessarily curative [31]. The objective is to use ISVRT as the prime modality for in situ vaccine neoantigen/neoepitope “extraction” repeatedly, adapting to the evolution of subclone repopulations as a priming dose before each cycle of IMT combinations. Other local combination therapies, like nanomedicines, intralesional drugs, etc., are known to induce in situ vaccination responses and can widen the intensity of the scope of neoantigen generation, along with ISVRT. New criteria for time–dose–fractionation for standard tissue tolerance may evolve following conventional knowledge and accumulated experience.

### 8.1. The Indications

For cancers with primary/disseminated metastatic disease not responding or progressing on IMT+/− CT/targeted therapy, the objective of the ISVRT schedule proposed here fulfills the dual purpose of inducing a local response (both direct and immunological) and a consistent, predictable abscopal effect. The expansion of further indications depends on the preclinical and clinical trials of the proposed approach.

### 8.2. Incorporate Antiangiogenics for Vascular Normalization

To improve the effects of ISVRT, the normalization of the vasculature is the first step to reduce hypoxia and sensitize cancer cells to radiation. Presently, one of the best-known vascular normalization groups of drugs is antiangiogenics, the effects of which usually last 28 days with the regular dose, called the “normalization window”. After this window, on the continuation of antiangiogenics, there will be an excessive/extensive pruning of blood vessels, a thickened pericyte, an increased deposition in the ECM, and the development of alternative angiogenic pathways such as vessel co-option with progressive worsening drug-resistant hypoxia (less likely to happen with low-dose schedules). These changes are reversible, as seen in patients who developed toxicities when drug holidays were given [35]. The reintroduction of the same or similar antiangiogenics reestablishes vascular normalization, making a case for the cyclical administration of AAGs to maintain the normalization of the tumor vasculature consistently. Theoretically, this cycle can be repeated several times. Anti-VEGF drugs can also lead to the restoration of vessel patency with improved blood flow, repairing pericytes with a baseline phenotype, etc., within seven days. Regression after stopping the drug is restored upon restarting the drug [85]. Another approach is giving low-dose AAGs continuously, especially multitarget TKIs, where the excessive thickness of pericyte coverage and other hypoxia-generating changes are avoided, leading to better oxygen diffusion and IMT combination drug deliveries [35,86]. In a trial, low-dose multitarget TKIs, through causing durable tumor vascular normalization and intratumoral CD4+ T, CD8+ T, and NK cell infiltrations, independent of interferon γ, improved the results of anti-PD-1 therapy with fewer side effects. Through their multitarget nature, these TKIs’ effects can encompass VEGFR, the fibroblastic growth factor receptor (FGFR), the platelet-derived growth factor receptor (PDGFR), and proto-oncogene tyrosine-protein kinase KIT (c-kit) [87].

### 8.3. ISVRT Pulsed Schedule for In Situ Vaccine Effect

SBRT (unlike conventional RT), with its enhanced cancer cell lysis mechanisms, generates significant amounts of neoantigens (sparing the normal tissues significantly). SBRT also induces an immune resistance “recoil” through at least four crucial pathways: (a). vascular disruption when the SBRT dose is >10 Gy per fraction [13], (b). Treg cell proliferation and activation [73], (c). two peaks of BMDC mobilization, one immediately and the other after two weeks [13]; and (d). the activation of the TREX1 pathway [29]. Overall, a dose per fraction of ≤10 Gy per fraction of ISVRT mitigates almost all these immune resistance recoil pathways and forms one of the foundational bases of the proposed strategy. A study indicates that even a single immunogenic radiation dose may be sufficient for a local response [31], supporting the ISVRT approach with a multiplier effect.

In patients with disseminated metastases, since the technique is explicitly aimed at “in situ vaccine generation”, one could select safer locations of metastases away from critical areas for ISVRT delivery, partial volumes [84], and low-dose regions near vital structures. The effects of the in situ vaccination of SBRT may be diversified, intensified, amplified, and sustained indefinitely (with activation of memory T cells) by cyclical ISVRT, keeping the total equivalent dose within acceptable toxicity levels by adopting the titration principle.

The parallel benefit of the ISVRT schedule is optimizing the not-so-inexhaustible RT dose by not delivering it during the hypoxic phase (or in hypoxic regions within the tumor) but timing it with each improved degree of oxygenation that is expected with the ongoing tumor response during the ISVRT course. Also, this approach can enhance innate immune mechanisms by decreasing the tumor burden and increasing the number of lymphocytes, possibly arming the memory cells against future mutations, recurrence, and second malignancies.

The radiotherapy routine dose fractionation schedule has taught us that cells become resistant if the overall time is prolonged beyond a specific period. SBRT can create both antigenicity and adjuvanticity [29]. Hence, the ISVRT proposed in the present article is primarily used to optimize the in situ vaccination effect. Therefore, considering the time gap in the ISVRT approach, redesigning the time–dose–fractionation (TDF) model is necessary to arrive at a maximum total dose that can be used for the ISVRT technique.

Theoretically, ablative doses of SBRT (>12 Gy dose per fraction) can cause “fibrotic nidus” with dead cancer cells, which can release neoantigens for a long time and potentially enhance memory cells continuously. Yet, underlying liver fibrosis and consequent fibrovascular remodeling impair both antigen recognition and the trafficking of immune cells [78]. Thus, keeping the ISVRT dose per fraction in the vascular–endothelial sparing range of 6 to 10 Gy per fraction is a harbinger of a supple ESM with maximum cancer cell lysis.

### 8.4. Integrating IMT

IMT can sensitize tumors for RT [88]. Thus, with the proposed schedule synchronizing the proliferating cancer cells to susceptible cell cycle phases, RT, CT, and IMT sensitize each other. Additionally, with the ISVRT approach, the interval of 3 weeks between each fraction allows radiation treatment volume reduction and dose modifications depending on the interim reduction in size, if any. Regarding the therapy combinations of the pulsed ISVRT schedule and CAR-T cells [75], the appropriate dose per fraction is yet to be decided.

IMT combinations or targeted therapies can be continued after the completion of the ISVRT course, as per the protocol for that particular type of cancer. Lesions no longer responding to IMT combinations may be resensitized to their original state after ISVRT’s in situ vaccine effect. In the KEYNOTE-01 trial in non-small-cell lung cancer, even at a median of 9.5 months after RT, the IMT group with a previous history of radiotherapy had longer OS and PFS [77], indicating the continued inherent role of an immunosupportive TME and memory cells.

### 8.5. ISVRT Immunological Manipulations

The mutation prevalent de novo, which the primary and booster effects of RT can release, and the antitumor effect can be expanded disproportionately with systemic or local immune adjuvants. This approach obviates the need to identify and harvest all neoantigens as and when they develop, enhances the synergy of SBRT and IMT when combined, minimizes immune escape, and reduces toxicities, including financial burdens. Another advantage of ISVRT is the dynamic matching with the evolved mutations de novo.

In the present day, the understanding of RT is undergoing a sea change, with irradiated tumors becoming an immunogenic hub for cancer elimination and RT acting as an efficient in situ vaccine generator and a systemic disease modifier to reject metastases. Successful RT provides lifetime immunological memory (cryptic vaccine) and dormancy with the restoration of immune editing [5], deterring immune escape. The immune enhancement effects of RT can be harnessed effectively by correcting the simultaneous mitigation of immunosuppressive “recoil”. In IMT-resistant patients, RT can restore the immune response and be amenable to more complex IMT manipulations [5].

### 8.6. ISVRT Immune Adjuvants Combinations

RT is usually synergistic and rarely, if at all, antagonistic with other forms of therapy as long as there are no overlapping significant toxicities. Therefore, as long as vascular integrity is maintained, ISVRT and different local therapy combinations and immunotherapies are potentially synergistic. In combination with other in situ vaccine producers, antigenicity and adjuvanticity promoters require systematic clinical trials around the primary paths discussed above. Nanomaterials impact every step of the immunity cycle [65] and can potentially improve the in situ vaccine mechanisms of ISVRT. The appropriate addition of immune adjuvants matching the innate immune mechanisms in humans will facilitate the creation of a memory cell bank.

### 8.7. ISVRT versus Trunk and Branch Mutations

Individual tumor trunk and heterogeneous branch mutations are present in all regions or at least two areas of the tumor, respectively, whereas private branch mutations are unique to one part [89]. In a study, branch and private branch mutations representing the intratumor heterogeneity of resistant subclones comprised about 40% of hepatocellular carcinoma [90]. RT enhances the response to IMT by targeting the trunk mutations [91]. Nanomedicines can potentially bridge the gap in all steps of in situ vaccination strategies [92]. According to the present author, analyzing and forming comprehensive combinations for the constantly shifting trunk and branch mutational patterns can be obviated by ISVRT delivered in divided doses, targeting the prevailing trunk mutations just before each cycle of IMT combinations without excluding other potential in situ local treatments or in vitro approaches.

### 8.8. Sequencing Surgery

The initiation of immune memory cells by SBRT and IMT is advantageous, and it did not happen with surgery in an animal model [66], although the removal of the tumor mass itself should increase the immunogenic surge. Surgery can be sequenced after in situ immunization treatments (akin to the total neoadjuvant protocol in rectum carcinoma) to maximize the initiation of the in situ effect and remove resistant cancer cell clones through surgery.

### 8.9. Criticality of ECM Suppleness

The algorithm in Figure 3 is used for TGFβ blockers [82] or adopting embryonic stem cell reversal techniques to normalize a treated area [93] to maintain the suppleness of the ECM. In a mouse model, in addition to dramatically expanding the cDC and pDC populations in the liver, an Flt3L-induced DC expansion resulted in metalloproteinase (MMP)-9-dependent fibrosis regression [78]. The angiotensin II receptor antagonist Losartan, with known antifibrotic activity, can deplete dense collagen networks via the TGF-β1 pathway, improve ECM functionality, and further enhance the efficacy of nanotherapeutics [94].

### 8.10. ISVRT Countering Accelerated Repopulation

Although no clear-cut conclusions can be drawn, the concomitant radiation boost delivered as the second fraction of the day starting from the 3rd week to handle the “accelerated repopulation” of the irradiated tumor showed an improved response trend in head and neck cancers. During this phase, cancer cells with resistant cancer phenotypes and stem cells repopulate and enter a relatively sensitive phase of the cell cycle [57]. The impact of the proposed pulsed ISVRT second dose, coinciding with the third week, is worth documenting.

### 8.11. Novel Unconventional and Other Radiotherapy Techniques

Also, in certain situations, partial volume treatment, targeting selectively the hypoxic segment of bulky tumors [84]; lattice therapy with a three-dimensional spatially arranged high-dose volume inside the tumor [95]; tumor core volume boosting [96]; multi-shell integrated boost techniques [97]; and other novel unconventional RT techniques [98] can be exercised as options for large tumors or those with a resistant residual volume after initial fractions/sessions. High-linear-energy-transfer (LET) radiation techniques, like the use of proton and carbon ions, have higher immunogenicity, increased stem cell clearance capability, improved immunological memory, and a higher sensitivity to hypoxia cancer cells [99]. FLASH radiotherapy is the other type of radiotherapy that one can look forward to in view of its ability to spare the vasculature better, with higher T-cell infiltration to the TME [100]. One can expect in situ vaccination applications of these modalities to come under the same principles as pulsed ISVRT, like photon therapy.

## 9. Summary and Limitations of the Article

Therapeutic SBRT is a course delivered in divided fractions, mostly over consecutive days. Chemotherapy and immunotherapy are administered in cycles, and RT was not tried on such a schedule. Animal trial results are already available using the pulsed RT concept. There is a theoretical possibility that radioresistant cell repopulation can occur during the intervals of ISVRT doses, even though each dose level proposed is adequate by itself immunologically [31], as well as a possibility of an effective pain control in bone metastases [34]. Although gaps in the proposed ISVRT may theoretically encourage an increase in metastases, mouse models have shown that pulsed RT in combination with immunotherapy shows fewer lung metastases and an increase in survival [19].

Resistance develops through several molecular mechanisms within cancer cells: A. Mutations in the existing pathways; B. rewiring bypassing the blocked pathways; and C. invoking new evolutionary pathways. Therefore, to overcome these changes, the proposed cyclical ISVRT strategy facilitates the new treatment approach in combination with existing drugs/methods. ISVRT avoids repeated testing to identify evolving clones with manifestations of resistance and keeps the option open to using related in vitro/intratumoral approaches. SBRT causes dose-related immunosuppression, e.g., the TREX1 pathway, and long-term hypoxia if the vasculature-endothelium is disrupted. Recent preclinical studies validate the importance of repeated antigen stimulations through the ISVRT approach of SBRT [19,20,21].

In conclusion, several critical conditions for cyclical/intermittent single-dose ISVRT are 1. overcoming the hypoxia immune-suppressive environment is the sine qua non for a successful in situ vaccination approach; 2. generating significant neoantigens/neoepitopes for a dynamic/adoptive in situ vaccination effect; 3. expanding the repertoire and repeated stimulations–booster vaccine effects and eliciting and enhancing innate immune memory cell pathways; 4. using appropriate immune adjuvants (including nanomedicines) to enhance the in situ vaccination effect, as in standard vaccination techniques against infections; 5. countering the “immunosuppressive recoil” of “concomitant immune tolerance” demonstrated by the tumor that are outside the field of radiation [101]; 6. expanding the four-dimensional effect of the number of neoantigens, widening the spectrum of antigens in different mutational types, intensifying antigenicity, and adapting to the trunk, branch, and private mutational evolution over time. Both the quantity and density of activated CD8+ cells [76] matter in the immune-therapy-induced response. Since the algorithm is complex and multifactorial, using the multiarm–multistage (MAMS) clinical trials method may be necessary, where multiple experimental treatments tested against a control arm within the same trial have several advantages in terms of time and resource consumption compared to when conducted as separate trials [102].

The proposed hypothesis fulfills several fundamentals of in situ vaccination requirements: The delivery of an endothelial-sparing 6–10 Gy dose of SBRT maximizes antigen generation, keeping both the short-term normalization of the vasculature and the long-term suppleness of the ECM conditions satisfied. Above all, the ISVRT schedule fulfills one of the critical requirements of any vaccination approach: the repetition of immunity cycles (moving the “wheel of immunity cycle”) and continued antigen presentations delivering the booster effects (Figure 2). To reiterate, fitting into the fundamental principles of vaccination, antigens, when presented repeatedly, unleash adaptive immune responses [30]. ISVRT, chemotherapy, and IMT, including CAR-T cells, can all instill enduring self-renewable memory T-cells [103]. The preprint version of the present article is documented [104]. Steps that optimize SBRT in the overall context of cancer reversal and the need for vascular normalization as a central theme of the research were published by Swamy K. earlier [11].

## Figures and Tables

**Figure 1 vaccines-12-00007-f001:**
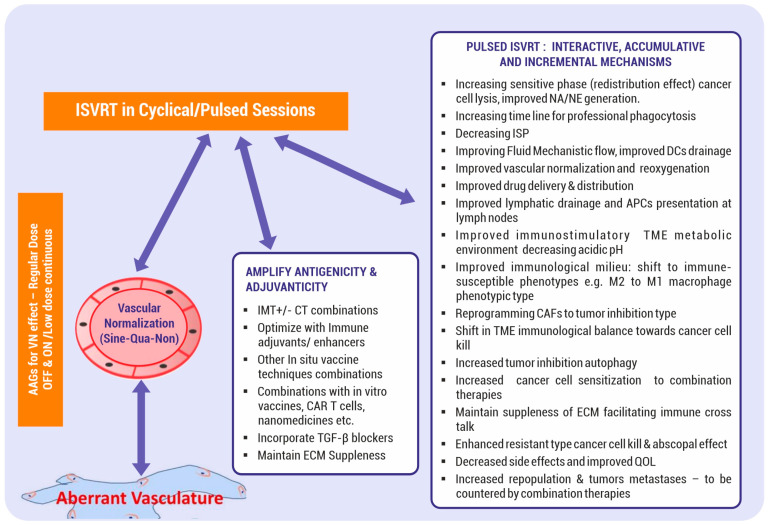
Pulsed In Situ Vaccination Radiotherapy (ISVRT) components. Fundamental (sine qua non) requirements: vascular, lymphatic vessel functions, and fibroblast normalization. Methods to improve antigenicity and adjuvanticity are listed in the middle text box. Pulsed RT results in a series of cumulative incremental benefits, as listed. VN = vascular normalization; AAGs = antiangiogenics; IMT = IMT; CT = CT; +/− = with or without; CAR T = chimeric antigen receptor; TGF-β = transforming growth factor beta; ISP = interstitial pressure; NA = neoantigen; NE = neoepitope; TME = tumor microenvironment; DCs = dendritic cells; APCs = antigen-presenting cells; CAFs: cancer-associated fibroblasts; ECM = extracellular matrix; QOL = quality of life.

**Figure 2 vaccines-12-00007-f002:**
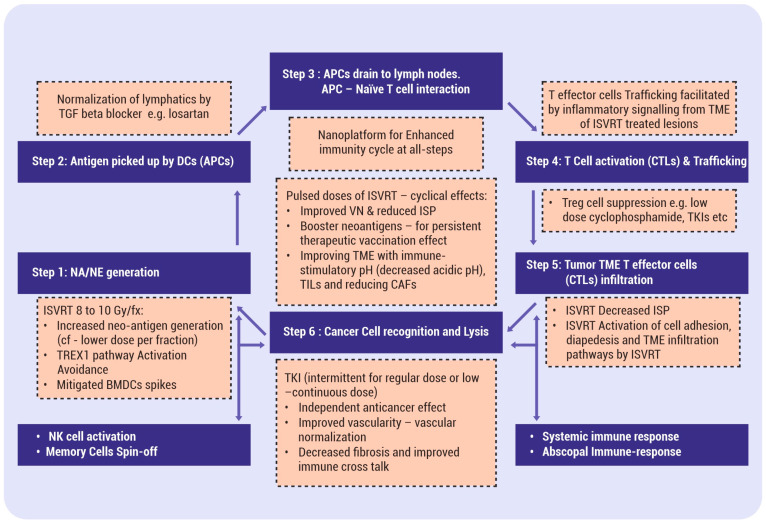
Stereotactic body radiotherapy (SBRT) and immunity cycle: The figure depicts the impact of SBRT on the immunity cycle at multiple steps and its integration with other immunity enhancers. Blue boxes: steps of immunity cycle. Orange boxes (with dashed lines): pathways to be integrated with ISVRT effects. ISVRT = in situ vaccination RT; NA/NE = neoantigen/neoepitopes; TREX1 gene; TGF beta = transforming growth factor-β; VN = vascular normalization; ISP = interstitial pressure; APCs = antigen-presenting cells; CAFs = cancer-associated fibroblasts; ECM = Extracellular Matrix; QOL = Quality of Life; AAGs = Antiangiogenics; IMT = IMT; CT = CT; CAR T = chimeric antigen receptor T; DCs = dendritic cells; T cells = T lymphocytes; CTLs = cytotoxic lymphocytes; TILs = tumor-infiltrating lymphocytes.

**Figure 3 vaccines-12-00007-f003:**
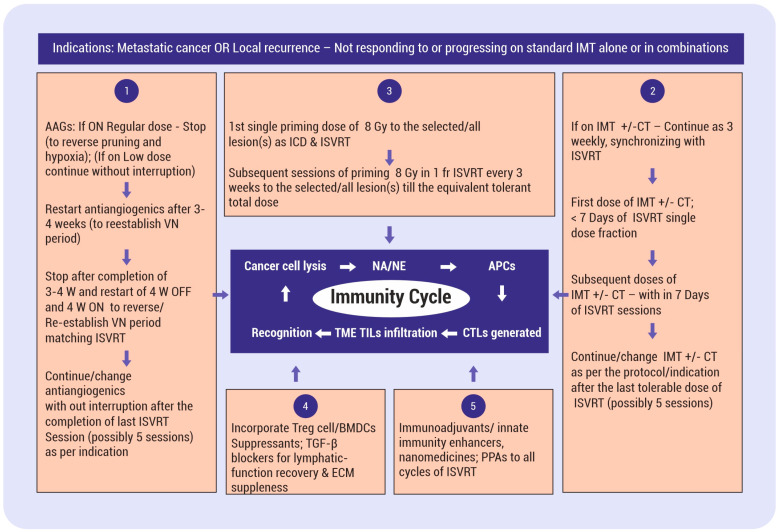
Algorithm for validating the proposed hypothesis using a pulsed/cyclical in situ vaccination stereotactic body radiotherapy dose (ISVRT). Numbers 1 to 5 are the essential components of the strategy integrating various therapeutic vaccine modulators. No. 1—antiangiogenics for vascular normalization. No. 2—immunotherapy +/− chemotherapy for synergism. No. 3—radiotherapy as a foundation for triggering the in situ vaccination effect. Subsequent doses of ISVRT synchronized with IMT or its combinations may act as a “boost” to reinforce the vaccine effect, which is fundamental to any vaccination program. No. 4—overcoming ISVRT immunosuppression pathways. No. 5—supplementing immune adjuvants, nanomaterials, other in situ technologies, professional phagocytic agents, etc. ISVRT = in situ vaccination stereotactic body radiotherapy schedule; W = week; AAGs = antiangiogenics; ICD = immunogenic cell death; VN = vascular normalization; TME = tumor microenvironment; APCs = antigen-presenting cells; ICD = immunogenic cell death.; NA = neoantigen; NE = neoepitope; BMDCs = bone-marrow-derived cells; Treg cell = T regulatory cell; TILs = tumor-infiltrating lymphocytes; IMT = immunotherapy; CT = chemotherapy; +/− = With or without; ECM = extra cellular matrix; TGF-β = transforming growth factor-beta; PPAs = professional phagocytic agents.

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
