# Peer review of "Therapeutic In Situ Cancer Vaccine Using Pulsed Stereotactic Body Radiotherapy—A Translational Model"

_vaccines, 2023, doi:10.3390/vaccines12010007_

Round 1
Reviewer 1 Report
Comments and Suggestions for Authors
The authors discuss the historical development of cancer therapeutic research, specifically in the context of radiation and cancer vaccines. It highlights that radiation's immunological potential may not have been fully realized. Cancer vaccines have evolved from being perceived as a failure in 2000 to a strategy with variable outcomes in 2023. The author introduces a new concept called "in-situ vaccination (ISVRT)" that involves using pulsed radiotherapy to generate tumor-specific neoantigens and epitopes, boost immunity, and synchronize with the vascular-normalized-immunity-cycle. This concept aims to optimize antigenicity and adjuvanticity for improved cancer treatment. The indications for this approach are in non-responding, locally progressing/metastatic cancer cases, and it can be combined with other therapies like immunotherapy, targeted therapy, chemotherapy, antiangiogenics, and vascular-lymphatic normalization. The hypothesis can be validated using a presented algorithm. Overall, the paper suggests a novel approach to cancer treatment through radiation therapy and immunological mechanisms.
Any role of macrophages in cancer vaccines? Do the authors discuss that in the text?
The figures could be of better resolution.
Author Response
The authors discuss the historical development of cancer therapeutic research, specifically in the context of radiation and cancer vaccines. It highlights that radiation’s immunological potential may not have been fully realized. Cancer vaccines have evolved from being perceived as a failure in 2000 to a strategy with variable outcomes in 2023. The author introduces a new concept called “in-situ vaccination (ISVRT)” that involves using pulsed radiotherapy to generate tumor-specific neoantigens and epitopes, boost immunity, and synchronize with the vascular-normalized-immunity-cycle. This concept aims to optimize antigenicity and adjuvanticity for improved cancer treatment. The indications for this approach are in non-responding, locally progressing/metastatic cancer cases, and it can be combined with other therapies like immunotherapy, targeted therapy, chemotherapy, antiangiogenics, and vascular-lymphatic normalization. The hypothesis can be validated using a presented algorithm. Overall, the paper suggests a novel approach to cancer treatment through radiation therapy and immunological mechanisms.
- Any role of macrophages in cancer vaccines? Do the authors discuss that in the text?
Ans: Yes, criticality macrophages in therapeutic in situ vaccination is discussed (appears in about 10 places, along with 1 reference).
- The figures could be of better resolution.
Ans: The text font size in the figures increased for better readability. I incorporated better resolution.
Please note that the title is refined to fit into the central proposed theme of the hypothesis and translational findings of animal studies.
Reviewer 2 Report
Comments and Suggestions for Authors
Dear colleagues,
In this manuscript, the author demonstrates a combinatorial translational hypothesis based on recent pulsed radiotherapy animal experiments and balancing the stereotactic body radiotherapies complex radioimmunological cascade. The article is devoted to personal author hypothesis which foundered by literature source. The figures reflect the results of the study but for supporting hypothesis could be expanded. Despite not bad impression of the article, there are necessary improvements in the structure of the manuscript. English should be corrected also.
In summary, I have been interested in the topic of the article. I believe this manuscript will attract attention from the research community. In my personal opinion, the article is valuable, a great prospect for further research, and, after significant corrections, can be recommended for publication.
Comments on the Quality of English LanguageDear colleagues,
In this manuscript, the author demonstrates a combinatorial translational hypothesis based on recent pulsed radiotherapy animal experiments and balancing the stereotactic body radiotherapies complex radioimmunological cascade. The article is devoted to personal author hypothesis which foundered by literature source. The figures reflect the results of the study but for supporting hypothesis could be expanded. Despite not bad impression of the article, there are necessary improvements in the structure of the manuscript. English should be corrected also.
In summary, I have been interested in the topic of the article. I believe this manuscript will attract attention from the research community. In my personal opinion, the article is valuable, a great prospect for further research, and, after significant corrections, can be recommended for publication.
Author Response
Dear colleagues,
In this manuscript, the author demonstrates a combinatorial translational hypothesis based on recent pulsed radiotherapy animal experiments and balancing the stereotactic body radiotherapies complex radioimmunological cascade. The article is devoted to personal author hypothesis which foundered by literature source.
- The figures reflect the results of the study but for supporting hypothesis could be expanded.
- Despite not bad impression of the article, there are necessary improvements in the structure of the manuscript. English should be corrected also.
In summary, I have been interested in the topic of the article. I believe this manuscript will attract attention from the research community. In my personal opinion, the article is valuable, a great prospect for further research, and, after significant corrections, can be recommended for publication.
Ans: Figures expanded. Figure 1 is split into two. Figure 1 depicts the factors associated with ISVRT/SBRT. Figure 2 illustrates the immunity cycle and pathways to be integrated with RT. Figure 3 (earlier figure 2) is modified for clarity. The flow of the article is restructured. MDPI Professional language editing is done.
Please note that the title is refined to fit into the central proposed theme of the hypothesis and translational findings of animal studies.
Reviewer 3 Report
Comments and Suggestions for Authors
I have read the interesting manuscript by Kumara Swamy on “Adaptive Conformable Therapeutic In-situ Cancer Vaccine Pathways – A Translational Model for Overcoming the Limitations”.
Below my comments:
- The preprint version of this article presents a clearer logic of the discussed arguments. Please better select, divide and elaborate the points in figure 1 and 2.
- Please introduce the separated parts A, B and C and justify why you focused on those three.
- Figure 1 and especially Figure 2, which is central for the hypothesis of the manuscript, has to be reformatted. They are very difficult to read. Moreover, figure 1 is not well described in part C.
Author Response
Reviwer 3
I have read the interesting manuscript by Kumara Swamy on “Adaptive Conformable Therapeutic In-situ Cancer Vaccine Pathways – A Translational Model for Overcoming the Limitations”.
Below my comments:
- The preprint version of this article presents a clearer logic of the discussed arguments.
Ans: Restructured for a better flow. Preprint flow is incorporated (Point 1 is about the hypotheses; Point 2 is about the impact of SBRT in all steps of the immunity cycle that need to be exploited; Point 3 is about overcoming the immunosuppressive cascade of SBRT; Point 4 about SBRT modulation of subclinical disease; Point 5 about: Antigenicity and adjuvanticity….. enhancement so on.
-Please better select, divide and elaborate the points in Figures 1 and 2.
Ans: Figures expanded. Figure 1 is split into two. Figure 1 depicts the factors associated with ISVRT/SBRT. Figure 2 illustrates the immunity cycle and pathways to be integrated with RT. Figure 3 (earlier figure 2) is modified for clarity.
- Please introduce the separated parts A, B and C and justify why you focused on those three.
Ans: A, B and C sections are replaced by continuous flow numbers. The logic, as given above, was adopted from the preprint version.
- Figure 1 and especially Figure 2, which is central for the hypothesis of the manuscript, has to be reformatted. They are very difficult to read. Moreover, figure 1 is not well described in part C.
Ans: Figures 1 and 2 are reformatted to 3 figures with explanations in the text and figure legends.
MDPI Professional language editing is done.
Please note that the title is refined to fit into the central proposed theme of the hypothesis and translational findings of animal studies.
Reviewer 4 Report
Comments and Suggestions for Authors
This manuscript provides a review of stereotactic body radiotherapies (SBRT) and raised the hypothesis that pulsed/intermittent/cyclical, endothelial-sparing single dose, in-situ vaccination (ISVRT) schedule as the in-situ vaccination shows distinct immunological induction. Overall, this review is interesting with a theme of radioimmunology. Although this review shows many information, it is neither clear nor well-organized for publication. First, there some information errors and inconsistency. At line 10, author wrote the failure is in 2000, while at line 41 it happens in 2020. At line 14, the “evolve” word is misused; it is not “author evolved concept”. Instead, author can say I propose the concept of…”. Interestingly, the “evolve” use is correct at line 60. At line 61, the “in-vivo” is not written correctly, it should be italic “in vivo” without the dash. At line 71, the content format is not right. Second, clarity is a problem in this review. Author likes to use long sentences that confuse our readers if the author himself is not confused. For example, at line 76-80, one sentence contains 63 words, while at line 779-784, there are 82 words in one sentence. Please split long sentences into short ones and do not use “, and” to connect many sentences (problem can be seen at line 33-34, line 37, and line 45, and line 771). Third, scientific review should have precise information and proper writing, while author needs to polish such parts in manuscript more intensively. For instance, at line 9-10, “impression” cannot be a “strategy”. At line 9, author can just write “is possibly not yet exploited completely” as “likely underexplored” for clarity. At line 11, author can directly show your hypothesis, instead of “hypothesis based on…”. Taken together, this review involves lots of information about SBRT and ISVRT with the theme of radiation and vaccine. However, it should be written with clarity, simplicity, and coherence.
Comments on the Quality of English LanguageEnglish writing is not clear enough for publication.
Author Response
Reviwer 4
This manuscript provides a review of stereotactic body radiotherapies (SBRT) and raised the hypothesis that pulsed/intermittent/cyclical, endothelial-sparing single dose, in-situ vaccination (ISVRT) schedule as the in-situ vaccination shows distinct immunological induction. Overall, this review is interesting with a theme of radioimmunology.
- Although this review shows many information, it is neither clear nor well-organized for publication.
Ans: The organization of the article is revised (taking into account the reviewer 3 comments as well). Figures were split (now 3 figures), redone, and explained further to improve the clarity of the text. The text flow primarily follows the steps of the immunity cycle (with pre- and post-factors). In order to match the flow of text, Figure 2 is designed to reflect the factors impacting every step of the immunity cycle and the pathways to be integrated. MDPI Professional language editing is done.
- First, there some information errors and inconsistency. At line 10, author wrote the failure was in 2000, while at line 41 it happened in 2020.
Ans: Corrected - 2020 to 2000 in line 41
- At line 14, the “evolve” word is misused; it is not “author evolved concept”. Instead, author can say I propose the concept of…”. Interestingly, the “evolve” use is correct in line 60.
Ans: Changed to - the present author is proposing the concept of ….
- In line 61, the “in-vivo” is not written correctly, it should be italic “in vivo” without the dash.
Ans: Corrected – replaced by in vivo in all 6 places
- At line 71, the content format is not right. Second, clarity is a problem in this review.
Ans: changed to “- The current review and hypothesis paper specifically proposes SBRT as an in vivo anticancer vaccination tool when used intermittently/cyclically in a pulsed manner, along with other immune modulators.
- The author likes to use long sentences that confuse our readers if the author himself is not confused. For example, at line 76-80, one sentence contains 63 words, while at line 779-784, there are 82 words in one sentence. Please split long sentences into short ones and do not use “, and” to connect many sentences (problem can be seen at line 33-34, line 37, and line 45, and line 771).
Ans: Line 76-80, 779-784, 33-34, 37, 45 modified, 771 – modified. Long sentences corrected. Also, the English editing of the document is done by MDPI professional editors.
- Third, scientific review should have precise information and proper writing, while author needs to polish such parts in manuscript more intensively. For instance, at line 9-10, “impression” cannot be a “strategy”. At line 9, author can just write “is possibly not yet exploited completely” as “likely underexplored” for clarity. At line 11, author can directly show your hypothesis, instead of “hypothesis based on…”.
Ans: Lines 9-10 were revised, and the word “impression” was removed. “Is possibly not yet exploited completely” is replaced with “likely underexplored.” Line 11 sentence deleted.
- Taken together, this review involves lots of information about SBRT and ISVRT with the theme of radiation and vaccine. However, it should be written with clarity, simplicity, and coherence.
Ans: The flow is reformatted for clarity and coherence without the A, B, and C subsections. Primarily guided by the preprint format (Reviewer 3 comment - preprint is more logical)
- Comments on the Quality of English Language English writing is not clear enough for publication.
Ans: MDPI Professional editing done
Please note that the title is refined to fit into the central proposed theme of the hypothesis and translational findings of animal studies.
Round 2
Reviewer 2 Report
Comments and Suggestions for Authors
Dear colleagues,
In this manuscript, the author demonstrates a combinatorial translational hypothesis based on recent pulsed radiotherapy animal experiments and balancing the stereotactic body radiotherapies complex radioimmunological cascade. The article is devoted to personal author hypothesis which foundered by literature source. The topic is original or relevant in the field Significant work had been performed by the author with the initial manuscript with corrections and changes in text, figures, and sources. Despite good impression of the performed changes, there are necessary some improvement details.
The figures reflect the results of the study but some text is hard for reading there.
There is source 20 in references with two different articles.
In summary, I have been interested in the topic of the article. I believe this manuscript will attract attention from the research community. In my personal opinion, the article is valuable, a great prospect for further research, and, after significant corrections, can be recommended for publication.
Author Response
- The figures reflect the results of the study but some text is hard for reading there.
Ans: Revised the entire highlighted text extensively. Significant corrections were made. Figure legends revised.
Also, English editing was done by the MDPI team earlier.
- There is source 20 in references with two different articles.
Ans: Corrected